# The Unraveling Loyalty Model of Traditional Retail to Suppliers for Business Sustainability in the Digital Transformation Era: Insight from MSMEs in Indonesia

**Mujianto Mujianto [1],\*, Hartoyo Hartoyo [1],\*, Rita Nurmalina [2] and Eva Z. Yusuf [1]**

[1]   School of Business, IPB University, Jl. Raya Pajajaran, Bogor 16151, Indonesia
[2]   Department of Agribusiness, Faculty of Economic and Management, IPB University, Jl. Agatis, Campus IPB Dramaga, Bogor 16680, Indonesia
\*   Correspondence: mujianto@apps.ipb.ac.id (M.M.); hartoyo@apps.ipb.ac.id (H.H.)

**Abstract:** The development of fast-moving consumer goods (FMCGs) retail has demonstrated an evolution of buyer–seller interactions. In the era of digital transformation, FMCGs and micro, small, and medium enterprises (MSMEs) can easily use website applications to shop for various products from suppliers, make payments, and access a wider variety of products with more efficient delivery. However, empirical studies on loyalty drivers on B2B relationships for business sustainability in the retail industry have not received much attention. This research aimed to examine the factors that influence loyalty and analyze the mediating role of MSME loyalty by discussing a new conceptual framework designed based on the buyer–seller relationship theory and relational marketing. Data were collected from 500 owners or managers of FMCG retail stalls in various provinces in Indonesia and analyzed using structural equation modeling (SEM). The results showed that merchandising, website quality, commitment, and satisfaction have a positive effect on loyalty, as opposed to service quality and trust. There are also different roles in the mediating variables of trust, commitment, and satisfaction on retail store loyalty. These findings were useful for policymakers, managers, and practitioners to clarify the influence of service quality, merchandising website quality, and the role of relationship quality on loyalty in the era of digital transformation.

**Keywords:** business sustainability; buyer–seller relationship; digital transformation; FMCG MSMEs; loyalty model; relationship marketing; relationship quality

## 1. Introduction

The fast-moving consumer goods (FMCGs) retail industry is regarded as one of the most attractive establishments in the country, with a sales value that continues to grow. According to Fortuneidn [1], the annual market value of FMCGs grew by 8.8% and 5.9% in Q3 2020 and Q3 2021, respectively. Irrespective of the fact that it was slower in 2020, Indonesia's growth in the FMCG retail industry exceeds the total value globally by 3.6%. It is even higher in most Southeast Asian countries. Malaysia occupies the second position with a growth rate of 5.5%, followed by Vietnam and Thailand with only 3 and 5%, respectively. Indonesia's FMCG market growth value outperformed other Asian countries, such as South Korea, China, Saudi Arabia, and the United Arab Emirates, by 5.2%, 3.1%, 4.3%, and −7.1%. This projection is bound to remain strong, as most households allocate 20% of their expenditures for FMCG products [2].

According to Kojogiang and Jimmy [3], the digital transformation era in Indonesia is marked by the emergence of technology-based companies with various new business formats and diverse suppliers of merchandise to traditional retail stalls. The new retail hybrid concept helps traditional stalls to adapt to the digital world. These stalls can easily shop for various products through digital platforms and website applications from suppliers, payment systems, and logistics to a more efficient supply chain [4]. These include

Mitra Bukalapak, Gudang Ada, Mitra Tokopedia, Sirclo, Toko Pintar, GoToko, GrabKios, ULA, Pawoon, Wahyoo, SRC (Sampoerna Strategic Community), DRP (Djarum Retail Partnership), and GGSP Partners (Gudang Garam Strategic Partnership). The increasing number of suppliers dealing with traditional retail stalls in Indonesia has changed the business environment, leading to increasingly fierce competition. Suppliers are described as being successful, assuming they can serve their customers' needs and offer attractive benefits by building long-term relationships [5]. Rauyruen and Miller [6] stated that loyal customers tend to stay with a particular supplier, thereby rejecting other competitors. They also strongly affect the company's performance, leading to greater profitability. The supplier relationship with the retail shop is in the context of a B2B relationship because traditional retail stalls purchase it in large quantities and is not consumed by itself [7].

B2B marketing is dominated by in industry marketing or organizational marketing, with a smaller number of customers compared to the end consumer market but with greater purchasing power [7]. For sustainable business growth, it is important to build consumer loyalty toward products or services [8]. Due to intense competition in the business market, efforts are needed to implement customer retention strategies to maximize customer relationships with their suppliers [9]. The existence of customers who do not buy, customers whose purchase frequency decreases, and customers whose average purchase decreases indicates a problem in loyalty in business relationships [10].

In the B2B environment, suppliers need to understand the nature and circumstances of their customers because of the unique characteristics of customers, because business customers buy products and services in large quantities, so the task of retaining customers is very important to achieve sustainable corporate competitive advantage [11,12]. However, suppliers in B2B relationships face a number of challenges due to the complex nature of the market, increased competition, increased deregulation, the presence of technological convergence, the rapid development of the internet, as well as increasingly customized products and services [13]. Therefore, the research question is what are the factors that influence retail store loyalty to suppliers in the digital transformation era?

A key success factor for achieving competitive advantage in the retail industry is providing products or services that satisfy customers, because satisfied customers develop trust and commitment to suppliers and customers become loyal by making repeat purchases [9,14–17]. From various previous studies, it can be explained that the important factors influencing the success of relationship marketing-based strategies to build loyalty are market offering factors in the form of products and services, information technology factors, and relational factors in the form of trust, commitment, and satisfaction [18]. A number of determinants of customer loyalty have been identified in the academic literature. However, not many studies have explored the determinants of loyalty in B2B relationships, especially in the FMCG retail industry because previous customer loyalty research was mostly in the B2C business to consumer environment [19].

This study makes an important theoretical contribution to the customer loyalty literature by exploring the concept of antecedents in a B2B environment based on relationship marketing theory to fill the knowledge gap on customer loyalty in the retail industry. Building and maintaining loyalty in a B2B context is not easy. Building loyalty in B2B is complicated by the fact that the determinants of customer loyalty that underlie relational exchanges in a business context are not well understood. So, this research fills the knowledge gap by contributing to the development of substantive understanding regarding customer loyalty in the B2B environment, especially in the retail industry.

Incidentally, various theories were used to identify the factors influencing customer loyalty, such as trust and commitment [20] and relationship marketing theories [18,21]. Previous research has proven that customer loyalty is directly affected by trust and commitment [5,19,20,22]. Some studies also reported that the concept of customer satisfaction also affects their loyalty [6,19,23–25].

According Morgan and Hunt [20], relationship marketing is essential to foster better synergies to boost customer loyalty. Hunt et al. [18] studied the essential influences that

affect the success of a relationship marketing-based strategy, including market supply factors in the form of tangible and intangible attributes known as goods and services. Service quality is the main source of competitive advantage in the market [18]. Good quality service is an important prerequisite for optimal relationships between supply companies and customers. According to Haghkhah et al. [19], service quality, commitment, and trust significantly affect customer loyalty. Moreover, it affirms the mediating role of commitment and trust between service quality and customer loyalty. Market offering factors in tangible attributes, called goods, and also affects the successful strategy employed in relationship marketing [18]. One of the contributions of this research is the inclusion of tangible attributes, namely merchandising variables, in the relationship marketing framework. This affects customer satisfaction and loyalty [26,27]. These merchandising attributes include product diversity, availability, quality, and pricing strategies, which impact loyalty [26–30]. In a business-to-business relationship, its benefits are product quality, service support, delivery performance, supplier knowledge, time to market, and personal interaction. Meanwhile, relationship costs consist of the direct product (price) and process costs [30]. Cater and Cater [31] stated that product quality also positively affects loyalty. The empirical results obtained also proved that customer loyalty is negatively and positively affected by price and product quality, respectively [31].

According to Hunt et al. [18], information technology is another important concept that affects the success of relationship marketing. Chen et al. [32] stated that good website quality would positively impact customer trust, satisfaction, and loyalty to the company. It is important to note that for its implementation to encourage retail stall loyalty to suppliers, there is a need for driving attributes. These include website quality in terms of ordering applications that are easy to understand and use and provide clear information [21].

Several determinants of customer satisfaction and loyalty associated with web design quality have been extensively studied in B2C consumer relations and identified in the academic literature. Website quality is one such concept used in measuring website quality based on end user perception [21]. Good website quality is bound to have a positive impact on customer satisfaction [32,33]. Its existence will also impact customer confidence in the company [32,34,35]. Preliminary studies stated that good website quality tends to positively impact customer loyalty [32,34,36–38]. The relationship between website design quality and customer loyalty was briefly mentioned in the B2B marketing literature. Therefore, this research is aimed to fill the gap and obtain a better understanding of website quality and customer loyalty, especially in the current digital transformation era. The website quality variable was included by accommodating several literature reviews regarding relationships with B2C consumers.

This research was urgently carried out for several reasons. First, following the FMCG retail trade channel distribution, it was discovered that in 2020, traditional retail channels contributed the most, namely 67% of the total sales in Indonesia [39]. Second, its potential in traditional stores is still significant and plays an essential role in product distribution and economic turnover in Indonesia. Alenia.id [40] reported that the supply chain value of merchandise products was USD 58 billion, approximately IDR 817 trillion annually. Third, there are innumerable retail stalls in the country. According to data from the Ministry of Cooperatives and MSMEs, the number of traditional stalls in Indonesia is 3.6 million [41]. Fourth, several technology-based and payment service companies emerged to make traditional stalls new distribution channels with online-to-offline (O2O) initiatives. This involves the transformation of traditional retail stalls to modern types by providing digital services.

This paper is structured in such a way that the following sections after the introduction are used to review the literature on loyalty and business sustainability, digital transformation, service quality, merchandising, website quality, commitment, trust, and satisfaction literature. Following this literature review, in the next section we outline the research method and present the results of our analysis. It is then followed by a discussion of the main findings of this study. By discussing the profile of the respondents, the results of

the measurement model and the results of the analysis of the hypotheses. We conclude this paper by identifying some of the practical implications and theoretical contributions made by this research, and then we explain some of the limitations of this study and offer suggestions for future research.

## 2. Literature Review

### 2.1. Theoretical Background

2.1.1. Customer Loyalty and Business Sustainability

Developing customer loyalty is the main goal of an organization or company; customer loyalty is very important for business because it is an indication of business continuity, future revenue prospects, and because of business profitability [11,12]. Lam et al. [22] mentioned that it provides various benefits for the organization in the long run. However, in the context of B2B relationships, these advantages benefit both partners in the business [5]. Loyalty-related studies stated that supplier companies are advised to build mutually beneficial relationships with their customers [7]. It was further reported that adopting cooperative actions benefits both parties, increases competitiveness, and reduces transaction costs [22]. According to Lam et al. [6], customer loyalty has a significant positive effect on the profitability of B2B companies. Moreover, by staying with the same supplier and rejecting competitors, loyal customers provide a steady stream of income for the company [6]. Studies related to loyalty inform that supplier companies are advised to build mutually beneficial relationships between suppliers and customers [7].

In a B2B context, loyal customers are more likely to focus on long term profits and engage in cooperative relationships that are mutually beneficial to both partners, thereby increasing their competitiveness by further reducing transaction costs [42]. Companies should focus on retaining existing customers because the costs incurred to develop old customers are cheaper than acquiring new customers [43]. Chinomona and Sandada [44] state that the cost of developing new customers ranges from 5 to 9 times higher, compared to retaining existing customers. This shows that companies need to retain loyal customers because through cost-effectiveness because there is a high chance of survival and sustainable growth in the future [9].

Loyalty behavior is defined as a customer's desire to repurchase a product or service and to maintain relationships with suppliers or service providers [5]. Furthermore, customer loyalty is important in building and developing long-term, mutually beneficial relationships between suppliers and customers. This also involves the regular making of purchases at a specific company. Loyalty indicators are repeat purchases or loyalty to repeat purchases of products or services [6]. Purchase across by making purchases in other lines of products or services, referrals by recommending suppliers to friends or colleagues, and retention or resistance to negative influences on supplier companies [6]. According to Bricci et al. [10], there are four basic building blocks for organic growth based on customer loyalty: loyal customers buy and repurchase (purchase and repeat purchase), obtain additional referrals to buy more (buy more), recommend and bring in new customers (referrals), and give constructive feedback.

With efforts to meet the needs of business customers, many supplier and seller companies are involved in business–customer relations and see the importance of strategic management of supplier and customer relations [5]. A number of determinants of customer loyalty have been identified in the academic literature. Customer loyalty is further considered as a result of successful relationship marketing practices [43]. In this study, the authors identify the determinants of customer loyalty based on relationship marketing theory. The success of relationship marketing depends on the benefits that customers receive [18]. The impact of relationship marketing on customer loyalty depends on the functional values of relationship marketing that can be felt by customers on the quality of services, products, value and price position, added value, relationship quality, trust, and commitment. From various previous studies, it can be explained that the important factors influencing the

success of a relationship marketing based strategy are market offering factors consisting of products and services, information technology factors, and relational factors [18].

### 2.1.2. Digital Transformation of Traditional Retail Stalls in the FMCG

Bowersox et al. [45] defined digital transformation as a business reinventing process used to digitalize operations and formulate extended relationships in the supply chain; it is the radical use of technology to improve company performance [46]. According to Mazzone [47], digital transformation is a focused and continuous technological evolution of a company, business model, or methodology, both strategically and tactically.

New retail in the FMCG industry is a concept developed by a technology-based company to connect merchants and suppliers on online and offline shopping systems [4]. It is the concept of digitalizing traditional retail, which connects merchants and suppliers with technology by only targeting certain aspects, such as the ordering supply chain, payments, digital products, and shopping experiences [4]. According to Corry Anestia [48], the supply chain aspect of the order is associated with the provision of digital access for traders to connect with suppliers of FMCG products, and this is carried out by placing orders through supplier ordering applications to obtain more diverse product variants at affordable prices.

### 2.1.3. Relationship Marketing and the Buyer–Seller Relationship

Grönroos [49] defined relationship marketing as a promotional activity to establish, foster, and maintain mutual relations between the consumers and company, thereby sustaining the interests of each party. This concept is centered on a long-term relationship between producers and consumers [7]. According to Morgan and Hunt [20], relationship marketing includes all promotional activities to establish, expand, and maintain successful relational exchanges.

Hunt et al. [18] reported that firms engage in relational marketing because it enhances their competitiveness. In other words, it aids them in efficiently and effectively producing valuable market offerings for multiple segments [18]. This simply implies that they perform engagement when the relationship is perceived as some sort of resource. Therefore, relational resources can potentially improve a firm's market position and, in turn, its financial performance [18].

According to the buyer–seller theory, the goal of relationship marketing is to create, maintain, and enhance mutually beneficial relationships, with a clear and transparent focus between the company and its customers to achieve optimum results [7]. It is also important to note that in establishing business relationships between buyers, sellers and associates, there is a tendency to shift from the initial pattern of transactional marketing perspective to relational [7]. This is usually conducted with a shift from a product to customer orientation as a solution to enhance more dynamic relationships to achieve mutual benefits [50].

### 2.1.4. Service Quality

The service quality concept is considered one of the main constructs for building a successful customer relationship. Preliminary research has proven that in the context of B2B, it has a positive impact on relationship quality and customer loyalty [5,19,28,29,31,32,51–53]. In a long-term buyer–seller relationship, various service quality dimensions can affect the consumers' repurchasing intentions [54]. Therefore, service providers need to understand these dimensions and provide the best quality to boost customer satisfaction and loyalty [54]. Haghkhah et al. [19] stated that good service quality leads to customer loyalty, increasing trust and satisfaction with the company. Positive perceptions of this factor are believed to increase the chances of customer involvement in supporting the firm and developing loyalty [29].

The retail service quality scale (RSQS), according to Dabholkar et al. [55], is a measure of service quality in stores with use performance based measures, and their scale has sufficiently strong validity and reliability for capturing customer perceptions of store service quality retail. This research approach emphasizes retail businesses or daily necessities

stores. It is derived from the SERVQUAL development method [56] by modifying the dimensions in determining customer satisfaction in retail stalls. Identifying retail service quality dimensions paved the way for recognizing customers' perceptions of the services provided by retail outlets. Moreover, [55] carried out extensive research to develop the RSQS and identified five dimensions of retail service quality, namely physical aspects, reliability, personal interaction, problem solving, and policy. In this study, the authors used indicators from the RSQS, including reliability, personal interaction, problem solving, and policy.

### 2.1.5. Merchandising

According to Szymanski and Hise [57], merchandise attributes are defined as factors related to supply for sales. Specifically, these are defined as a variety of quality merchandise and products reflected by a certain price [29,31]. Moreover, it is an effort to procure products in the right quantity, availability, and variety [26,28,58]. Products can be a good starting point for satisfying and creating customer loyalty [29]. It is an expression of the behavior intended to support a product by communicating the experience associated with its usage to others and saying positive things about the item [29]. Based on the content analysis of online customer comments, [59] identified certain factors related to products that can affect customer satisfaction, namely competitive prices, merchandise availability, and conditions.

Internet-based retailers should be able to meet the competitive prices uploaded on the website [59]. However, merchandise condition also tends to affect customer satisfaction, and its availability should be displayed on the retailer's website [59]. Some other research further stated a positive relationship between product offerings and customer satisfaction in online shops [58]. According to Qin et al. [60], certain preliminary studies have also proven that selling quality products at low prices is not enough to satisfy longtime loyal customers, and a stronger foundation is needed to implement customer relationships concerning service quality [60]. The dimensions of merchandising are an assortment [26,28,58], price [27,31,59–61], product quality [29], and availability [59].

### 2.1.6. Website Quality

WebQual, a development of ServQual compiled by Parasuraman, was designed in 1998. It has undergone several iterations concerning varying dimensions and variables. Barnes and Vidgin [62] stated that the site quality analysis is categorized into three focus areas. They include site quality, quality of interaction offered by services, and information provided, better known as WebQual 3.0 [62].

Website quality is important in terms of attracting and retaining existing customers [32]. A high-quality website not only determines the customer's decision to make purchases, it is also the main reason for them to buy online products [36,63]. Relationship marketing theory states that any website with good quality is bound to have a positive impact, thereby affecting customers' loyalty to the company [32,33,35,36,38,63,64]. According to Barnes and Vidgin [62], its dimensions are information quality, which includes reliability, ease of understanding, relevance, and accuracy. Meanwhile, service interaction quality is easy to communicate and feeling safe completing personal data during monetary transactions. The overall impression, which is the total appearance of the site, is good [62].

### 2.1.7. Trust, Commitment, Satisfaction

Trust is a customer's feeling regarding whether or not the type of service rendered meets their hopes and expectations [65]. According to Mayer et al. [65], based on the perspective of relational marketing theory and the development of loyalty between companies and their customers, this attribute is extremely important. Morgan and Hunt [20] stated that in B2B, relational marketing is centered on building and developing trust. Understanding this attribute and its contribution to loyalty affects how B2B relationships are developed and managed between companies and their customers [20]. Several studies reported that trust is the key to successful relationship development in the B2B marketplace [5,19,32].

According to Parasuraman et al. [66], trust is the main construct that triggers the successful service relationship between the company and its customers. It exists when one party has confidence in the reliability and integrity of the exchange partner [20]. Meanwhile, Palto et al. [54] stated that trust is essential in creating customer loyalty and retention. Askarizad and Babakhani [67] reported that in the context of B2B relationships, it has a direct effect on loyalty.

Mayer et al. [65] defined trust as a person's willingness to be sensitive to others with the hope that these individuals take certain actions that impact people who believe in them. However, this is regardless of their ability to control and monitor them. According to Mayer et al. [65], it comprises three components, namely ability, benevolence, and integrity. The relationship marketing literature emphasizes that trust is an important variable in boosting customer loyalty. The previous study also supports its effect on loyalty, especially in B2B [5,19,32,51,68,69].

Rauyruen and Miller [5] argued that an individual or group's commitment to suppliers is crucial for developing customer loyalty. Interestingly, commitment and trust are the main components needed in building and maintaining long term relationships between business partners. These are essential for developing a successful relationship [20]. The commitment between a company and its customers has been defined as either an implicit or explicit bond, and this, in turn, leads to the development of long-term benefits. It is also the sincerity to keep an agreement based on the explicit and implicit willingness of the transaction partners (the company and its customers) to continue a mutually beneficial relationship [70].

Albari [71] stated that from the company's perspective, commitment helps to facilitate the tying of long-term relationships with its customers. However, this is usually proven through various customer loyalty activities. According to Mattila [72], the commitment that leads to customer loyalty consists of several dimensions, namely, affective or emotional, calculative or cognitive, and goal. Affective commitment reflects the customer's emotional attachment to the service provider, which is the company [72]. Then, calculative commitment is the customer's desire to maintain or continue the relationship with the company. This is based on their experiences because the company was able to meet their needs, thereby preventing the anticipation of terminating the relationship or the cost of switching to competing brands [72]. Goal commitment is occasionally indicated by determining customer activities toward certain objectives [72].

Oliver [73] stated that satisfaction is derived from all activities performed during the purchasing process and not only from observing or directly consuming products or services. This variable is important in maintaining purchase intention [5]. Customer satisfaction is the overall evaluation of a company's post purchase performance or service utilization. According to Parasuraman et al. [74], it mainly arises from a cognitive process in which the feeling of satisfaction is due to the comparison process between the company's perceived performance and customer's expectations. In relationship marketing theory, there is consistent evidence that satisfaction contributes to repurchasing and behavioral intentions, as well as customer retention and loyalty. Some research on business-to-business relationships has demonstrated the existence of a correlation between satisfaction and loyalty [5,6,19,22,24,25,31,75].

## 2.2. Empirical Studies Related to the Research Model Framework Using the SEM Model

In the past, multivariate regression methods and factor analysis were frequently used to detect interdependence in the research problem being examined [76]. Incidentally, in the last few decades, the structural equations model (SEM) application has been widespread and often used, especially in social sciences that study the behavior of various actors, such as consumers [77].

In the following section, from the various literature that has been described, hypotheses are developed to build the model framework in this study, as seen in Figure 1. The variables used are service quality, merchandising and website quality trust, commitment, and satisfaction to determine the effect on loyalty. Considering that in the era of digital

transformation, traditional retail stalls in their wholesale purchases to suppliers use ordering applications, the merchandising factor, whose indicators consist of assortment product, price, product quality, and product availability, needs to be further investigated for its effect on loyalty, as there are more and more suppliers. Service quality variable is certainly an interesting variable to study along with supplier reliability regarding delivery and problem solving, as well as several policies from suppliers, such as free shipping and payment methods in applications, such as late payments via Fintech. We also take the website quality factor as its own variable to determine its effect on loyalty in this study. Due to the offline to online digital transformation era, the interaction of personnel (employees of suppliers) has decreased somewhat and has shifted to applications as a means and tool for ordering wholesale goods to suppliers. The ease of the supplier ordering application certainly has an effect on retail stall loyalty, apart from the interactive display of the application and also the information contained in the ordering application, which is clear and informative.

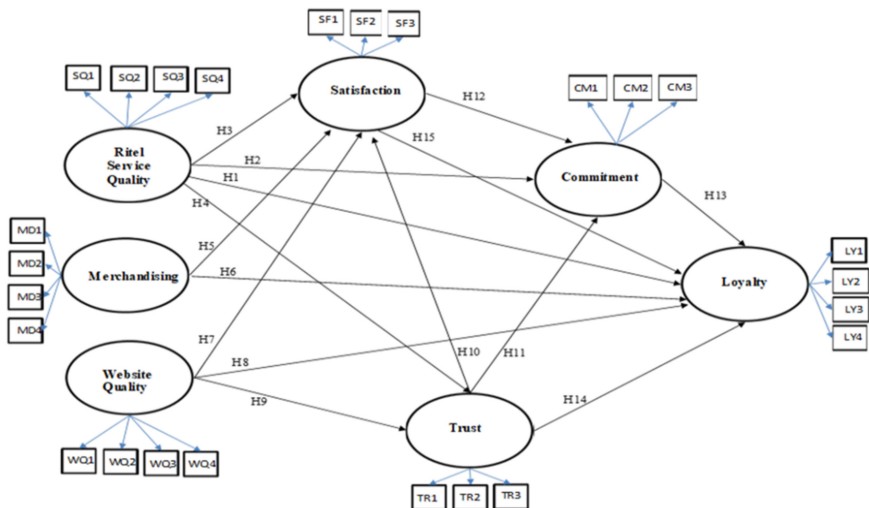

**Figure 1.** The conceptual framework model of the traditional retail stalls' loyalty to suppliers (developed by the researcher).

### 2.2.1. Service Quality Affects Customer Loyalty, Trust, Commitment, and Satisfaction

Haghkhah et al. [19] stated that service quality affects customer loyalty. Then [5], confirmed that it increases purchase intention and loyalty. This is in line with the research conducted by [51], stating that service quality positively affects customer loyalty in B2B relationships. However, different results were obtained by [25], reporting that neither satisfaction nor service quality directly affected loyalty attitudes or behavior in B2B relationships. Based on the outcome of previous studies, the first hypothesis (H1) was proposed as follows:

**Hypothesis 1 (H1).** *Service quality has a positive significant effect on customer loyalty.*

Relationship marketing theory supports the connection between service and quality. In addition, commitment and trust can be regarded as their components [19]. These results are consistent with relational marketing theory and previous research [5,51]. Palto et al. [54] stated that service quality strongly correlates with satisfaction and trust and is weakly related to commitment. The main advantage of relational marketing is that both the company and consumers benefit from the relationship, especially when trust is built by the organization through the delivery of quality services [68]. Chiou and Droge [25] further reiterated that interactive service quality results in total satisfaction and trust, while the facility type is a driver of both variables. Chennet et al. [78] illustrated that this attribute impacts trust and the importance of service differentiation in achieving a high level of commitment, satisfaction, and positive word-of-mouth. Service quality is weakly correlated

with commitment, and service quality is strongly correlated with trust [54]. According to Izogo and Ogba [68], service quality has the greatest level of positive impact on trust.

Chumpitaz and Paparoidamis [51], as well as Rauyruen and Miller [5], stated that customers are more loyal to businesses with higher service quality, thus leading to greater investment of trust and commitment, which are key components of a quality relationship. Haghkhah et al.'s [19] research findings in the Iranian automotive industry investigated the effect of service quality and customer value on trust and commitment. According to Hangkhan et al. [19], service quality has a positive and significant effect on trust and commitment. In addition, the effect of customer value on trust is positive and significant. Based on the literature and previous research, the second (H2) and third (H3) hypotheses were proposed as follows:

**Hypothesis 2 (H2).** *Service quality has a positive significant effect on commitment.*

**Hypothesis 3 (H3).** *Service quality has a positive significant effect on trust.*

Veloso et al. [23] reported that service quality has a significant effect and is the main determinant of customer satisfaction. This is in line with the empirical results obtained from previous research [32,33,52,54,59,79,80]. Kimani et al. [81] affirmed that retail service quality affects customer satisfaction in both supermarkets and stalls in Kenya. Wiputra et al. [82] further asserted that it affects customer satisfaction in traditional retail industries. Based on the literature and previous research, the fourth (H4) hypothesis was proposed as follows:

**Hypothesis 4 (H4).** *Service quality has a positive significant effect on satisfaction.*

### 2.2.2. Merchandising Affects Customer Satisfaction and Loyalty

Cater and Cater [31] stated that in a B2B relationship, satisfaction is negatively affected by price. It was further reported that product quality positively and negatively affects loyalty behavior and price. Different results were obtained by [29], conveying that, surprisingly, product quality had a negative impact on customer loyalty in B2C consumer marketing relationships. Based on previous research, the fifth hypothesis was proposed as follows:

**Hypothesis 5 (H5).** *Merchandising has a positive significant effect on the loyalty.*

Then, Mustaqimah et al. [27] confirmed that the product, price, and promotion affect customer satisfaction. According to Dharmesti and Nugroho [28], product variations have an insignificant effect on online satisfaction. This is not in line with the findings of [59], arguing that product diversity affects customer satisfaction with internet shopping. Interestingly, this was reinforced by the findings of [33], arguing that product variations affect customer satisfaction with online shops. The lower the cost of searching for information, the greater the product varieties, and the more attractive it appears to consumers. This is because they can obtain more information and select various products from a specific place. Based on previous research, the sixth hypothesis was proposed as follows:

**Hypothesis 6 (H6).** *Merchandising has a positive significant effect on satisfaction.*

### 2.2.3. Website Quality Affects Customer Satisfaction, Trust, and Loyalty

Website quality positively affects repurchase intention and customer satisfaction [34]. This is in line with the research by Jauhari et al. [36], stating that this attribute significantly affects satisfaction and consumer buying interest. Chen et al. [32] also reported that website quality positively affects customer satisfaction and e-loyalty in e-commerce systems. Moreover, Hsu et al. [79] stated that this attribute affects customers' pleasure, satisfaction, and purchase intention to engage in transactions.

Rasli et al. [63] reported that website design, information quality, transaction, and payment capabilities directly and positively impact customer satisfaction. This is in line with the findings from previous literature [32,34,35,37,38,79]. However, a different result was obtained by Noronha and Rao [83], stating that website design had a less significant effect on customer satisfaction. According to Hsu et al. [79], it is more important than information and system quality in influencing customer satisfaction and repurchase intention. This is in line with the finding that web design directly affects customer loyalty to online shops [28,84]. The research carried out by Chen et al. [32] proved that the system's quality and information on the website positively affected e-loyalty in e-commerce. It was further asserted that website design is essential and positively affects user trust, satisfaction, and loyalty. These results led to the proposition of the following three hypotheses, namely the seventh (H7), eighth (H8), and ninth (H9):

**Hypothesis 7 (H7).** *Website quality has a positive significant effect on customer satisfaction.*

**Hypothesis 8 (H8).** *Website quality has a positive significant effect on customer loyalty.*

**Hypothesis 9 (H9).** *Website quality has a positive significant effect on trust.*

### 2.2.4. Trust Affects Satisfaction and Commitment

According to Morgan and Hunt [20], trust and commitment are important variables in building a successful relationship. Furthermore, relationship marketing theory states that companies become involved in some form of relational exchange with suppliers if they believe its benefits outweigh the associated costs [30]. The theory built around relational marketing emphasizes the importance of trust as a fundamental necessity in successfully developing long-term relationship elements [10].

Ulaga and Eggert [30] also asserted that trust significantly affects commitment. This is in line with the empirical findings from other research [10,22,37,68,85]. However, it is contrary to Palto et al.'s [54] research that trust is weakly related to commitment. According to Samudro et al. [85], customer satisfaction develops trust in the relationship between suppliers and buyers. Furthermore, customer trust affects commitment. Chiou and Droge [25] reported that trust affects satisfaction. Bricci et al. [10] stated that it positively and directly affects satisfaction. The findings from previous literature led to the proposition of the tenth (H10) and eleventh hypotheses (H11):

**Hypothesis 10 (H10).** *Trust has a positive significant effect on satisfaction.*

**Hypothesis 11 (H11).** *Trust has a positive significant effect on commitment.*

### 2.2.5. Satisfaction Affects Commitment

Generally, satisfied customers are highly committed to a particular shop and are always happy to recommend it to friends and relatives [86]. It directly impacts customer commitment because higher levels are bound to affect purchase behavior repeatedly [37,87]. This is in line with the previous studies that satisfaction also affects commitment in the service industry [85,88]. Therefore, this led to the proposition of the eleventh hypothesis (H12), namely:

**Hypothesis 12 (H12).** *Satisfaction has a positive significant effect on commitment.*

### 2.2.6. Trust, Commitment, and Satisfaction Affect Customer Loyalty

Menidjel et al. [87] stated that trust has the greatest impact on behavioral loyalty because a higher level leads to absolute loyalty. Therefore, customers with a high level of trust are bound to be loyal. The increasing level of customer commitment tends to boost loyalty. The main reason for this phenomenon is that a high commitment level helps customers engage in long-term relationships. Rauyruen and Miller [5] confirmed

that to maintain customer loyalty, suppliers need to improve the four relationship aspects, namely trust, commitment, satisfaction, and service quality. The empirical results from the earlier mentioned study proved that suppliers seek to boost customer satisfaction to maintain behavioral loyalty with SME clients. This is in line with the previous research on B2B that satisfaction has a positive effect on loyalty [6,19,22,24,25,31,75] and in the B2C context [10,23,28,32,52,82,89,90].

Chiou and Droge [25] stated that consumer trust and satisfaction are the seeds of loyalty behavior. This is because they boost loyalty attitudes in high service product markets and directly or indirectly persuade consumers to invest in certain assets. Hannan et al. [75] further reported that satisfaction and trust directly affect customer loyalty. To create and maintain an SME customer loyalty attitude, suppliers need to build relationships based on trust and commitment and provide an excellent service system. This is consistent with the empirical findings of other research that trust affects customer loyalty in B2B [19,22,25,75] and B2C marketing relationships [32,37,89,91,92]. Rauyruen and Miller [5] also stated that commitment to suppliers is crucial for developing customer loyalty. Based on their findings, it is deduced that commitment affects customer loyalty. This is in line with the empirical findings from previous research on B2B [19,82] and B2C relationships [37,92,93]. These led to the proposition of the following three hypotheses relating to trust, commitment, and satisfaction with customer loyalty in H13, H14, and H15, as follows:

**Hypothesis 13 (H13).** *Commitment has a positive significant effect on customer loyalty.*

**Hypothesis 14 (H14).** *Trust has a positive significant effect on customer loyalty.*

**Hypothesis 15 (H15).** *Satisfaction has a positive significant effect on customer loyalty.*

### 3. Research Methodology

The purpose of this research is to identify and analyze the determinants of the loyalty of traditional retail stalls owners to suppliers in the digital transformation era. In addition, it also analyzes the role of relationship quality consisting of trust, commitment and satisfaction in the loyalty of traditional retail stalls to suppliers in the digital transformation era. We formulate the research questions as follows: (1) what factors influence the loyalty of traditional retail stalls to suppliers? (2) What is the role of trust, commitment, and satisfaction in the loyalty of traditional retail stalls to suppliers? To answer these questions, we adopt a survey method. We used survey research based on the perspective of traditional retail stall respondents as a buyer of products and services to suppliers. Data collection only used a cross-sectional approach, with responses measured using a Likert and semantic differential scale from one to five. Number one is used to express strongly disagree (very low), and five represents strongly agree (very high). As mentioned by Huang, and Benyoucef [94], survey methods are widely used to measure behavior change in research in the fields of social studies and e-commerce. Chin and Newsted [95] stated that the survey method is relevant for estimating behavior and relationships between variables [96].

### 3.1. Sample and Data Collectian

The population of respondents used in this study were the owners or managers of traditional retail stalls who had become members of Alfamikro Alfamart suppliers, totaling 60,612. Regarding the number of samples, this study follows the rule of thumb of the SEM tool, where the sample size is 5–10 times the number of indicators [76]. Because the number of indicators in this study is 57, the minimum number of samples must be 285.

Sampling was carried out using a proportional purposive sampling technique [97]. Researchers distributed questionnaires to 550 traditional retail stalls, with proportional distribution to Alfamikro Alfamart areas/branches. In this case, the researcher took samples in each area (branches) based on the proportion of the number of retail stalls in each region divided by the total number of retail stalls from those that have become supply partners.

Then, the incoming questionnaires were examined and re-checked, and incomplete questionnaire entries were not included in data processing. There are 500 samples that can be processed further after a cross check of the 550 respondents who entered with the amount per each area or branch from the Supplier, as shown in Table 1.

**Table 1.** Number of research samples of traditional retail stalls for Alfamikro suppliers.

| No | Region | Branch | Qty Retail | n |
|---|---|---|---|---|
| | | | | Traditional Retail Stalls |
| 1 | | Medan | 1.940 | 16 |
| 2 | | Pekanbaru | 3.393 | 28 |
| 3 | | Palembang | 1.091 | 9 |
| 4 | Sumatera | Batam | 364 | 3 |
| 5 | | Kotabumi | 740 | 6 |
| 6 | | Jambi | 728 | 6 |
| 7 | | Lampung | 727 | 6 |
| 8 | | Balaraja | 3.393 | 28 |
| 9 | Banten | Cikokol | 1.333 | 11 |
| 10 | | Serang | 2.666 | 22 |
| 11 | | Parung | 2.908 | 24 |
| 12 | Jabotabek | Bekasi | 1.212 | 10 |
| 13 | | Bogor | 4.847 | 40 |
| 14 | | Cileungsi | 1.939 | 16 |
| 15 | | Karawang | 1.575 | 13 |
| 16 | | Cianjur | 1.212 | 10 |
| 17 | West Java | Bandung | 2.424 | 20 |
| 18 | | Cirebon | 1.454 | 12 |
| 19 | | Cimahi | 1.213 | 10 |
| 20 | | Cilacap | 6.059 | 50 |
| 21 | Central Java | Rembang | 1.939 | 16 |
| 22 | | Klaten | 1.334 | 11 |
| 23 | | Semarang | 2.787 | 23 |
| 24 | | Sidoarjo | 2.668 | 22 |
| 25 | East Java | Malang | 2.908 | 24 |
| 26 | | Jember | 2.302 | 19 |
| 27 | Bali Nusa | Bali | 608 | 5 |
| 28 | | Lombok | 1.939 | 16 |
| 29 | Kalimantan | Pontianak | 735 | 6 |
| 30 | | Banjarmasin | 242 | 2 |
| 31 | Sulawesi | Makassar | 1.454 | 12 |
| 32 | | Manado | 486 | 4 |
| | Grand Total | | 60.612 | 500 |

Source: processed and developed by the researcher.

After knowing the number of proportions for each region, sampling was also based on the purposive sampling technique. Purposive sampling is a sampling technique with certain considerations with a random sampling methodology, where the sample group is targeted to have certain attributes [97]. The specific attributes or criteria in sampling in this study are (a) the owner or manager of a retail shop business who has become a supplier member of Alfamikro Alfamart in the retail cluster, (b) traditional retail stalls that sell their merchandise and already use the goods ordering application and (c) located in areas covered by Alfamikro Alfamart suppliers that are easy to reach for being respondents.

*3.2. Data Analysis*

The data analysis technique used to predict and confirm the relationship between the research variables and the owners or managers of traditional retail stalls is SEM (structural equation modeling), performed with the Lisrel 8.70 program. This method is commonly employed to evaluate statistical and causal models. It is also used to determine the struc-

tural relationship between similar multiple regression equations [76]. This approach aids in describing the correlation between constructs consisting of both dependent and independent variables involved in the analysis [76]. Then, multi-variable techniques are classified as interdependent or dependent. This indicates that SEM can be categorized as a unique combination due to its stance on factor analysis and multiple regression [76]. It is a statistical tool that simultaneously solves multilevel models that linear regression equations cannot solve. SEM is also regarded as a combination of regression and factor analyses. Moreover, it is dependent on covariance analysis to provide a more accurate matrix than linear regression.

The data validity is used to determine the suitability of each indicator while also evaluating its reliability. The most basic test tool is measuring the overall fit based on the likelihood ratio of the chi-square, which is considered sensitive to the sample size. However, a model is presumed to be good if the chi-square value is low. A smaller value indicates that the accepted model is better at *p* > 0.05. The model's fit with the index has to meet the criteria shown in Table 2 [76].

**Table 2.** Criteria for the goodness of fit index.

| Category | GOFI | Acceptable Level of Conformity |
|---|---|---|
| Absolute fit measures | Chi-square ($\chi^2$) | Expected small |
| | GFI | >0.9 |
| | RMSEA | <0.08 |
| | Standardized RMR | <0.5 |
| Incremental fit measures | AGFI | >0.9 |
| | CFI | >0.9 |
| | NNFI | >0.9 |
| | NFI | >0.9 |
| | RFI | >0.9 |
| | IFI | >0.9 |
| Parsimonious fit measures | PNFI | 0–1 |
| | PGFI | 0–1 |
| | Normed chi-square | 1.0–3.0 |

### 3.3. Variable Operations

According to [97], the operationalization of variables is a definition dependent on theory. However, these are presumed to be operational, assuming they can be measured or even tested. It provides information on how to measure the research variables.

These variables consist of several dimensions and indicators. Moreover, its operationalization process is shown in Table 3. Each indicator is measured using a Likert scale, with 1 = Strongly Disagree, 2 = Disagree, 3 = Doubtful (between agreeing and disagree), 4 = Agree, and 5 = Strongly Agree.

**Table 3.** Operationalization of variables.

| Variable | Variable Operational Definition | Indicator | References |
|---|---|---|---|
| Service Quality | Quality of retail service that suppliers render to their partners. | Reliability: The suppliers keep their promises and do things right to ensure their partners remain loyal. Personnel interaction: Employees who are polite and willing to help shop partners. Problem-solving: Supplier employees who can handle potential problems, such as customer complaints, returns, and exchanges. Policy: Operating hours, payment options, and free delivery from suppliers to partners. | [55] |

**Table 3.** *Cont.*

| Variable | Variable Operational Definition | Indicator | References |
|---|---|---|---|
| Merchandising | Procurement of products in the right quantity, variety, and price by suppliers. | Assortment (diversity) of products at suppliers. Price: The price of the product from the supplier to the stall partner. Durability or the expiry date of the product is good. Availability: the suppliers' products are always available. | [29,30,59] |
| Website Quality | The quality of the existing website based on user perception. | Usability: Usability (ease of use and ordering application, as well as an easy-to-understand website, due to its attractive appearance). Information quality: reliable, easy to understand, relevant, and accurate information. Service interaction quality: The service interaction quality triggers easy communication and customers feel safe while completing personal data and transactions. Overall Impression: The overall appearance of the site is good. | [62] |
| Trust | The trust of certain parties toward others in developing transactional relationships is based on the perception that the trusted individual fulfills all obligations as expected. | Ability: Ability refers to the competence and characteristics of suppliers in providing, serving, and securing transactions from others' interference. Benevolence: Kindness is the sellers' willingness to provide mutually beneficial satisfaction between themselves and the consumer. Integrity: Integrity relates to the behavior or habits of suppliers in going about their businesses. The information provided to consumers is accurate or factual, and quality products are sold. | [65] |
| Commitment | Seriousness in fulfilling certain agreements is based on the explicit and implicit willingness of the transaction partners, namely recipients and service providers. This is to continue the functional relationship that has been established. | Emotional commitment: Affective commitment reflects the consumer's emotional attachment to the service provider Cognitive commitment: Calculative commitment is a person's desire to maintain or continue a relationship with his partner based on the experience associated with meeting needs. Goal commitments: Goal commitment can be proven by occasionally determining individual activities toward certain objectives. | [71] |
| Satis faction | Consumers' overall evaluation of the supplier's post-purchase performance or service utilization. | Overall satisfaction with the products and services of the supplier. Satisfaction with the customer's ideal product or service. Considering the expectations of these stalls and the conditions provided by the supplier. | [98] |
| Customer loyalty | Conditions of customers' loyalty or consumers regularly making purchases. | Repeat purchases: Loyalty to repurchase products or services. Purchase across product and service lines: Purchase on another product or service line. Referrals: Supplier recommendations to friends or colleagues. Retention: Resistance to negative effects on the supplier company. | [6] |

Source: processed and developed by the researcher.

## 4. Results and Discussion

### 4.1. Respondent Profile

The respondent profiles are described based on gender, age, education, length of business, average turnover, and the informant's domicile, as shown in Table 4. A total of 500 respondents from various regions participated in this research. The majority, approxi-

mately 59.4%, were women, and 40.6% were men. This is based on the fact that women open retail stalls to earn more income for the household. To help improve the necessities of life, they need to be encouraged to work to impact the family economy positively. This is in line with the research carried out by [99], stating that women choose micro-enterprises due to gender inequalities in the labor market and flexibility of time, as well as economic opportunities in business. They tend to open micro-enterprises due to limited network capabilities and strategies. The low level of family income is also one of the reasons that cause them to engage in this activity [100].

**Table 4.** Respondent profile in this research.

| Description | Category | Qty | Total % |
|---|---|---|---|
| Gender | Man | 203 | 40.6 |
| | Woman | 297 | 59.4 |
| Age | 17–25 years old | 29 | 5.8 |
| | 26–35 years old | 153 | 30.6 |
| | 36–50 years old | 266 | 53.2 |
| | Over 50 years old | 52 | 10.4 |
| Education | <Junior High School and Junior High School | 86 | 17.2 |
| | Senior High School | 355 | 71 |
| | Association/Bachelor's degree | 58 | 11.6 |
| | Magister/Master's degree | 1 | 0.2 |
| Length of business | <1 year | 49 | 9.8 |
| | 1–5 years | 227 | 45.4 |
| | 6–10 years | 135 | 27 |
| | More than 11 years | 89 | 17.8 |
| Average turnover/day | <500 thousand | 63 | 12.6 |
| | 1–2 million | 128 | 25.6 |
| | 500 thousand–1 million | 192 | 38.4 |
| | More than 2 million | 117 | 23.4 |
| Domicile | Banten and DKI Jakarta | 135 | 27 |
| | West Java | 64 | 12.8 |
| | Central Java and Yogyakarta | 119 | 23.8 |
| | East Java | 65 | 13 |
| | Sumatera | 74 | 14.8 |
| | Bali Nusa Tenggara | 21 | 4.2 |
| | Kalimantan and Sulawesi | 22 | 4.4 |

Source: processed by the researcher.

In a micro-business, retail stalls do not require large capital or special skills. Regarding age, the most observed respondents, 53.2%, were within 36 to 50 years, while 30.6% were between 26 to 35 years. This reflects that the average owners or managers of retail stalls are in their productive age. Based on the aspect of education, they are dominated by high school graduates, approximately 71%, followed by lower than high school graduates with 17.2%. This shows that, on average, the owners or managers of retail stalls are not properly educated. Moreover, the majority, 45.4%, have been in business for one to five years. Those who have been trying to open their stalls for six to ten years are 27%. Regarding the average sales turnover, as many as 38.4% of the stall owners earn IDR 500 thousand to 1 million per day, while 25.6% earn one to two million. This shows that most of the respondents aged 36 to 50 years with high school educational qualifications are mostly women. They have been in this business for one to five years, making an average daily turnover of IDR 500 thousand to 1 million; additionally, most of them reside in Java.

### 4.2. Structural Model

The measurement analysis was continued with the structural model in accordance with the objectives. The main one provides a path diagram based on t-values and standard solutions. It was further tested with a statistically fair goodness of fit index (GOFI) value. Furthermore, the structural aspect is a model where the goodness of fit for the inner one is proven by testing the effect of each exogenous latent variable on the endogenous attributes.

The compatibility test of the main research model with the total sample is shown in Table 5. It was discovered that of the eleven GOFI model fit measures, seven results are good (good fit), two are close to good (marginal fit), and the remaining two are not good (close fit). Ref. [77] stated that the assessment of model fit is based on how many of its sizes can meet the cut-off value in the research. The more the target value of the match with the GOFI size is met, the better the results. Based on the data in Table 5, with a much higher number of matches, it was concluded that the overall model has a good GOFI.

**Table 5.** Analysis of the overall model's goodness of fit test.

| GOF | Cutoff Value | Result Value | Description |
|---|---|---|---|
| *Chi*-square ($\chi^2$) | Preferably smaller than Df | 663.9 | Marginal Fit |
| Probability (*p*-value) | $\geq$0.05 | 0.000 | Marginal Fit |
| RMR | Good models have a small RMR $\leq$0.05 atau 0.08 Hair et al. [76] | 0.0778 | Good Fit |
| RMSEA | $\leq$0.08 | 0.073 | Good Fit |
| GFI | $\geq$0.90 | 0.849 | Marginal Fit |
| AGFI | $\geq$0.90 | 0.809 | Marginal Fit |
| CFI | $\geq$0.90 | 0.992 | Good Fit |
| NFI | $\geq$0.90 | 0.988 | Good Fit |
| NNFI | $\geq$0.90 | 0.991 | Good Fit |
| RFI | $\geq$0.90 | 0.986 | Good Fit |
| IFI | $\geq$0.90 | 0.992 | Good Fit |

Table 6 shows that the SLF value of each variable has fulfilled the goodness of fit model requirements. Therefore, the service quality (SQ), merchandising (MD), website quality (WQ), trust (TR), satisfaction (SF), and customer loyalty (LY) are valid. This is also supported by a t-value of 1.64 (real level 10%), which indicates these variables are significant. The overall model has good construct reliability with CR and VE of 99.7% and 92.6%, respectively. These results indicate that the variables have met the standard provisions or have been declared valid. CR and VE are declared valid if the values are >70% and 50%, respectively. Furthermore, based on Figure 2, the magnitude of the effect between variables is significant.

According to Hair et al. [76], the measurement of a construct on reliability and validity can be used to ascertain whether each indicator can explain each of its latent variables. A construct can have good validity for the construct or its latent variable if it fulfills the SLF (standardized loading factor) requirement of >0.70. A construct can be relied upon if it has a construct reliability CR (construct reliability) of greater than or equal to 0.7 and the value of construct validity is VE (variance extractive) and below 0.5. Construct reliability is a measure of the internal consistency of the indicators of a formed variable that shows the degree of the formed variable, while the variance extracted is a measure of how much total variance of the indicators is extracted by the formed variables. CR and VE testing was carried out to test the reliability value of the measurement model for each latent variable.

**Table 6.** SLF values and t-value models.

| Variable Let | Indicator | Coefficient/ SLF (λ) | T-Value (≥1.64) | Error Var | λ² | Reliability CR ≥ 0.7 | VE ≥ 0.5 | Descr. |
|---|---|---|---|---|---|---|---|---|
| | | | | | | 0.99 | 0.961 | Good |
| Service Quality (SQ) | (SQ1) | 0.98 | 23.71 | 0.0396 | 0.96 | | | |
| | (SQ2) | 0.98 | 23.9 | 0.0396 | 0.96 | | | |
| | (SQ3) | 1 | 24.44 | 0 | 1 | | | |
| | (SQ4) | 0.96 | 22.46 | 0.0784 | 0.922 | | | |
| | | | | | | 0.943 | 0.806 | Good |
| Merchandising (MD) | (MD1) | 0.9 | 20.03 | 0.19 | 0.81 | | | |
| | (MD2) | 0.99 | 24.1 | 0.0199 | 0.980 | | | |
| | (MD3) | 0.73 | 14.6 | 0.4671 | 0.533 | | | |
| | (MD4) | 0.95 | 22.01 | 0.0975 | 0.903 | | | |
| | | | | | | 0.989 | 0.956 | Good |
| Website Quality (WQ) | (WQ1) | 0.93 | 21.52 | 0.1351 | 0.865 | | | |
| | (WQ2) | 1 | 24.31 | 0 | 1 | | | |
| | (WQ3) | 1 | 24.36 | 0 | 1 | | | |
| | (WQ4) | 0.98 | 23.7 | 0.0396 | 0.96 | | | |
| | | | | | | 0.989 | 0.967 | Good |
| Trust (TR) | (TR1) | 0.98 | | 0.0396 | 0.96 | | | |
| | (TR2) | 0.99 | 68.81 | 0.0199 | 0.98 | | | |
| | (TR3) | 0.98 | 55.22 | 0.0396 | 0.96 | | | |
| | | | | | | 0.986 | 0.96 | Good |
| Commitment (CM) | (CM1) | 0.98 | | 0.0396 | 0.96 | | | |
| | (CM2) | 0.99 | 60.56 | 0.0199 | 0.98 | | | |
| | (CM3) | 0.97 | 51.67 | 0.0591 | 0.941 | | | |
| | | | | | | 0.991 | 0.974 | Good |
| Satisfaction (SF) | (SF1) | 0.98 | | 0.0396 | 0.96 | | | |
| | (SF2) | 0.99 | 65.01 | 0.0199 | 0.98 | | | |
| | (SF3) | 0.99 | 66.79 | 0.0199 | 0.98 | | | |
| | | | | | | 0.97 | 0.89 | Good |
| Customer Loyalty (LY) | (LY1) | 0.96 | | 0.0784 | 0.922 | | | |
| | (LY2) | 0.99 | 55.32 | 0.0199 | 0.98 | | | |
| | (LY3) | 0.94 | 36.58 | 0.1164 | 0.884 | | | |
| | (LY4) | 0.88 | 28.73 | 0.2256 | 0.774 | | | |
| | **Overall model CR 99.7%** | | | | | | | |
| | **Overall model VE 92.6%** | | | | | | | |

Source: processed and developed by the researcher.

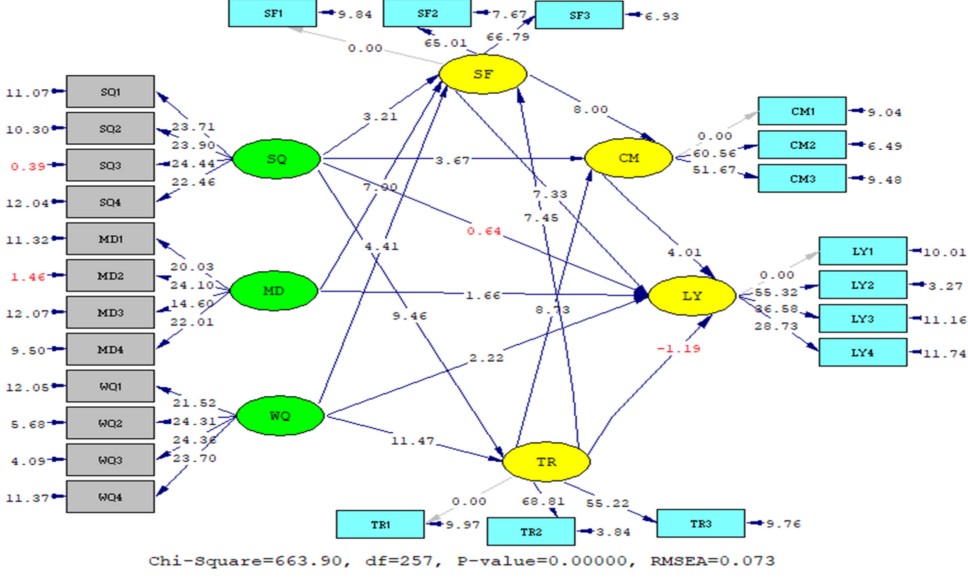

**Figure 2.** Output t-value based on SEM with the Lisrel 8.80 Model 1. Source: processed by the author.

The t-value explanation regarding the output in each research dimension is shown in Figure 2. Subsequently, the path diagram or image consists of retail service and website qualities, merchandising, trust, commitment, satisfaction, and loyalty. Based on Figure 2, information on the t-value and path coefficients that connect the research variables directly or indirectly are obtained. Table 7 shows fifteen relationships, with thirteen being significant and positive and two insignificant.

**Table 7.** Hypothesis testing results.

| Hypothesis | Line (Relationship) | Value t-Count (≥1.64) | Effect | | Hypothesis Conclusion |
|---|---|---|---|---|---|
| | | | Direct | Total | |
| H1 | SQ→LY | 0.64 | 0.03 | 0.03 | Rejected |
| H2 | SQ→CM | 3.67 | 0.15 | 0.15 | Accepted |
| H3 | SQ→SF | 3.21 | 0.14 | 0.14 | Accepted |
| H4 | SQ→TR | 9.46 | 0.44 | 0.44 | Accepted |
| H5 | MD→SF | 7.00 | 0.28 | 0.28 | Accepted |
| H6 | MD→LY | 1.66 | 0.06 | 0.06 | Accepted |
| H7 | WQ→SF | 4.41 | 0.19 | 0.19 | Accepted |
| H8 | WQ→LY | 2.22 | 0.13 | 0.13 | Accepted |
| H9 | WQ→TR | 11.47 | 0.52 | 0.52 | Accepted |
| H10 | TR→SF | 7.45 | 0.41 | 0.41 | Accepted |
| H11 | TR→CM | 8.73 | 0.46 | 0.46 | Accepted |
| H12 | SF→CM | 8.00 | 0.38 | 0.38 | Accepted |
| H13 | CM→LY | 4.01 | 0.30 | 0.30 | Accepted |
| H14 | TR→LY | −1.19 | −0.09 | −0.09 | Rejected |
| H15 | SF→LY | 7.33 | 0.53 | 0.53 | Accepted |
| | **Total Effect** | | **4.76** | | |

Source: processed and developed by the author.

### 4.3. Qualitative Analysis of Retail Stalls Loyalty Models to Suppliers

4.3.1. The effect of Service Quality on Loyalty, Commitment, Satisfaction, and Trust

Based on the results of the analysis, the retail service quality has an insignificant effect on loyalty with a t-value of 0.64, signifying it is <1.64 (real level 10%). Therefore, hypothesis 1 is rejected. Generally, good service quality leads to customer loyalty, which triggers trust and satisfaction with the company or supplier. The supplier's service quality orientation can increase retail stall loyalty. This research indicates that retail service quality does not positively affect customer loyalty. The relationship between satisfaction and commitment fully mediates the effect of service quality on customer loyalty. In other words, this variable only affects customer loyalty through relational quality, namely satisfaction and commitment.

These results align with the research by [69], which states that service quality has an insignificant effect on customer loyalty. However, the relationship quality fully mediates the effect of service quality on customer loyalty. Ref. [80] also obtained a similar result, stating that service quality has a negative relationship and an insignificant effect on loyalty.

This research is inconsistent with the findings of [19], which states that service quality has a strong and positive effect on customer loyalty. Complex business environments, especially B2B, require a high level of service quality from their suppliers in return for customer loyalty. Incidentally, this finding is also not in line with previous research [5,51], which states that service quality significantly affects customer loyalty.

The results of this research indicate that quality services are not enough to create loyal customers; in this case, retail stalls. However, it is when the quality of service provided by the suppliers and long-term relationships will show signs of loyalty from the retail stalls. This is important in the digital transformation era. With intense competition and many suppliers offering similar services to retail stalls, developing successful and mutually beneficial relationships with them is necessary to boost loyalty. The results also show that the retail service quality variable affects commitment because the t-value is SQàCM:

3.67 > 1.64 (real level 10%). Therefore, hypothesis 2 is accepted, implying that each increase in retail service quality by one unit triggers commitment by 0.15.

Commitment is a behavioral element that aids to maintain a long-term relationship between the two parties, thereby strengthening the connection between suppliers and retail stalls. The results of this research indicate that retail service quality affects commitment. Unfortunately, being loyal to suppliers cannot be directly achieved by service quality. This variable needs to be improved to trigger trust and commitment in retail stalls.

These results are in line with the research carried out by [19], which states that service quality has a positive and significant effect on trust and commitment. Ref. [78] also stated that this variable positively affects commitment. The importance of service quality in achieving a high level of commitment ultimately leads to satisfaction and positive word-of-mouth promotion. Ref. [68] also stated that companies with good service quality can build strong relationships with their customers because this variable positively affects trust, and commitment. However, the findings of this research are inconsistent with [54], in which informed service quality weakly correlates with commitment.

Commitment is one of the most essential and central concepts for understanding the strength of customer relationships. It is also useful for measuring loyalty and predicting customers' repeat buying behavior. Suppliers' provision of quality service significantly affects the commitment of retail stalls. Therefore, they are committed to shopping from wholesale suppliers for their merchandise.

The analysis results also show that the retail service quality variable affects satisfaction due to the t-value of SQàSF: 3.21 > 1.64 (real level 10). Therefore, hypothesis 3 is accepted. An increase in retail service quality by one unit boosts satisfaction by 0.14 units. The higher the quality of services provided by suppliers, the greater the satisfaction. This encourages them to develop and maintain relationships with their customers. These results are in line with some research that service quality has a positive relationship and a significant effect on customer satisfaction [23,32,52,54,78–80,82,101–104].

The existence of quality services, especially in terms of appropriate support, knowledge of suppliers in terms of problem-solving and personal interaction, certainly encourages retail stall satisfaction. This is in line with the findings of [31], which states that in the B2B context, satisfaction is positively affected by delivery performance, supplier knowledge, and personal interaction.

This research proved that the following variables reliability, personnel interaction, problem-solving, and policy simultaneously significantly affected the satisfaction of retail stalls. These results are also supported by [82], stating that the quality of retail services, including reliability, personnel interaction, as well as problem-solving and policy, has a simultaneous effect (together) on satisfaction. It was also proven that this variable is a determining factor in the satisfaction of retail stalls with suppliers and indirectly affects their loyalty. The analysis showed that the retail service quality influences trust because the t-value is SQàTR: 9.46 > 1.64 (real level 10%). Therefore, hypothesis 4 is accepted, implying that for each increase in retail service quality by one unit, confidence is boosted by 0.42 units.

Retail stalls' trust is the spearhead in a business relationship with suppliers. The best service is based on the belief that the supplier can also provide quality products. There is a relationship between these variables because the better the service provided by the supplier, the greater the trust. This depends on the supplier due to the guarantee of good service quality. In the retail distribution industry, customers purchase the promised goods, not tangible ones, and they trust the service provider to ensure a sustainable relationship. Moreover, in such circumstances, service quality helps build trust and loyalty by cultivating certain relationships. Retail stalls could argue that building trustworthy relationships requires efficient and effective delivery of core products and services, as well as ensures supplier honesty, reliability, and integrity in dealing with them. There is absolute trust if the wholesale goods ordered can be delivered at the appropriate time and in an orderly manner. The existence of cash on delivery payment policy encourages trust in suppliers.

These results are in line with the research carried out by [54], which states that service quality strongly correlates with satisfaction and trust, even though it does not have a similar strength effect. In addition, it has a positive and significant effect on trust. These are also consistent with some previous research [19,25,32,68,78].

4.3.2. The Effect of Merchandising on Satisfaction and Loyalty

The results of the analysis show that the merchandising variable affects satisfaction because the t-value is MDàSF: 7 > 1.64 (significant level 10%). Therefore, hypothesis 5 is accepted, which indicates each increase in merchandising by one unit is bound to boost satisfaction by 0.24 units. Quality products are a good starting point to encourage customer loyalty [29]. Merchandising is the most critical component in implementing supplier strategic decisions related to goods diversity, availability, quality, and pricing strategy offered to retail stalls.

The diverse and various merchandise are the main attraction for retail stalls. The ability to manage its choice with the appropriate amount is an important factor for suppliers. They need not purchase varying merchandise with excessive amounts, for the slow-moving nature of goods burdens inventory costs. Moreover, the availability of the goods should not be allowed to go out of stock. This is because it causes retail stalls to be dissatisfied, looking for other suppliers or sources to meet their needs. On the other hand, if the goods sought for are not available, it will also reduce the potential sales value. There is also a need to consider the supplier's determination of the price strategy because this variable also affects the satisfaction of retail stalls.

Retail stalls are bound to be satisfied with dealing with suppliers if they provide a variety of goods needed for wholesale shopping. However, when placing an order, the goods sought must be available, undamaged, and worthy of sale at competitive prices. This is defined as the price level that retail stalls obtain from suppliers, and it is easier to resell the items to the end users, thereby making a reasonable profit.

These results align with the research conducted by [26], which states that merchandising positively and significantly affects customer satisfaction. Ref. [58] further product diversity affects customer satisfaction regarding learning to use the internet. According to [33], merchandise attributes positively affect online shopping satisfaction. Ref. [59] also reported that its condition affects customer satisfaction. Moreover, merchandise availability should also be displayed on the retailer's or supplier's website during online shopping because it impacts customer satisfaction. Ref. [27] stated that marketing mix variables such as product, price, and promotion affect consumer satisfaction.

However, this research does not support the findings of [31], which states that satisfaction is negatively affected by price, while product quality has an insignificant effect on customer satisfaction. Ref. [6] also stated that price negatively affects satisfaction. Then, [60] reported that selling quality products at low prices is not enough to satisfy longtime loyal customers. Some other studies reported that another relationship role associated with service is needed. The analysis also showed that merchandising affects loyalty because the t-value of MDàLY: 1.66 > 1.64 (real level 10%) is 0.06 units; thus, hypothesis 6 is accepted.

The availability of various quality products has not been able to encourage retail stalls to repurchase suppliers. These stalls still have many options for making repeated purchases from other suppliers. Based on these results, the merchandising variable can cause customers to become loyal. Suppliers need to have a better understanding of the wants and needs of these stalls. They also need to pay attention to their product assortment, continuous availability, quality, and selling price. These findings are not in line with the research carried out by [31], which states that price loyalty behavior is negatively affected. Ref. [29] also stated that, surprisingly, product quality does not positively affect customer loyalty.

Some research has conveyed innumerable supporting attributes related to these findings, such as merchandising effects on loyalty. Ref. [28] reported that product variations significantly affect customer loyalty in the e-commerce market. Ref. [61] further stated

that it positively and significantly affects customer loyalty in the retail industry. Furthermore, [31] reported that product quality positively affects loyalty behavior. These are strengthened by [26], which states that merchandising positively impacts customer loyalty. It is easier for companies to apply merchandise as a strategy to attract and retain customers. Supplier's ability to provide various products with high quality, in terms of expiration, worth, and availability, tends to ultimately affect the satisfactory behavior of retail stalls and loyalty to suppliers.

4.3.3. The Effect of Website Quality on Satisfaction, Trust, and Loyalty

The results showed that the website quality affects satisfaction because the t-value is WQàSF: 4.41 > 1.64 (real level 10%). Therefore, hypothesis 7 is accepted, indicating that every increase in website quality by one unit tends to boost satisfaction by 0.24 units.

One of the supporting factors that can trigger consumer interest is an e-commerce website. Its existence presents quick and precise information to customers without engaging in marketing through print media first. Surprisingly, any website that provides extensive information in an easy-to-digest format, well-designed navigation, and smooth operation can boost consumer satisfaction. This is due to the feeling that its activities are considered effective.

The rapid growth of the internet has a reflective impact on marketing. This encourages entrepreneurs to adopt e-commerce as a medium for interacting and performing transactions with customers. To attract retail stalls, suppliers need to design their websites according to their needs and based on the placed orders and transactions made. Online shopping applications are expected to help them practically place orders for wholesale goods. There is no need to close the stalls or incur additional costs by visiting the supplier's place because the shop's opening hours are more productive. Moreover, website service quality has emerged as an important factor that positively correlates with customer satisfaction, as well as encourages visiting and revisiting. These results align with the research by [34], which states the importance of online-based companies or suppliers to identify the website service quality factors with the most positive effect on customer satisfaction. Ref. [63] stated that website design, information quality, transaction, and payment capabilities directly and positively impact customer satisfaction. Ref. [36] also reported that website quality significantly affects consumer satisfaction.

Suppliers should be able to create an easy-to-understand website and utilize product ordering applications. The understanding and use of information on this platform and website navigation are factors that ensure retail stall owners or managers are satisfied. The analysis results show that the website quality affects loyalty because the t-value is WQàLY: 2.22 > 1.64 (real level 10%). Therefore, hypothesis 8 is accepted, implying that each increase in website quality by one unit triggers loyalty by 0.13.

These results align with the research by [36], which states that website quality significantly affects customer loyalty. Ref. [32] also reported that the web system's quality positively impacts e-loyalty and e-commerce. Similar results were also reported by [101]. It was further reported that e-retail supply chain companies should manage the quality of their websites in terms of product prices, attractive catalogs, accessibility, and content design, thereby making it easy for customers to understand and engage in repurchase activities.

Suppliers dealing with retail stalls in this current digital transformation era should be able to design easily understood goods ordering applications with quality websites. Information uploaded on the ordering application needs to be clear, such as prices, stock availability, and ease of placing orders, which are essential factors that cause retail stalls to be loyal to suppliers. This application should be user-friendly considering that, on average, retail stall owners or managers are age not millennials. Therefore, a system that is easy to understand and use is needed to place orders and engage in wholesale shopping transactions.

The results also show that the website quality variable affects trust because the t-value is WQàTR: 11.47 > 1.64 (real level 10%). Therefore, hypothesis 9 is accepted. As such,

increasing website quality by one unit boosts trust by 0.54 units. This is because trust is the main factor that affects how to shop on online sites. Without this variable, any business in the online system will not run smoothly. The trustworthiness of a retail stall depends on the information the supplier uploads on the website. Trust is a key factor that makes these shops engage in online purchases from suppliers.

Ref. [35] stated that website quality variables, such as user interface quality (UIQ), information quality (IQ), perceived security risk (PSR), and perceived privacy, affect satisfaction and trust. Good website quality is bound to affect trust, increasing customer buying interest. Ref. [32] also reported that information uploaded on the website positively affects trust in e-commerce systems. Notably, this finding is in line with the results of subsequent research [32,34,35].

### 4.3.4. The Effect of Trust on Satisfaction, Commitment, and Loyalty

The analysis results show that trust affects satisfaction because the t-value is TRàSF: 7.45 > 1.64 (real level 10%). Therefore, hypothesis 10 is accepted, implying that every increase in confidence by one unit triggers satisfaction by 0.41 units.

Trust is the basis or foundation of a business, and it is also an awareness or customers' perception of a product. Service providers use this tool to establish long-term relationships and boost customer satisfaction. The effect of this variable provides a sense of consumer satisfaction in terms of using the stipulated product or service. The theory built around relational marketing emphasizes the importance of trust as the foundation and fundamentals needed to develop elements of long-term relationships to implement this strategy successfully.

These results are in line with [10], which reports that trust has a positive and direct effect on satisfaction. This variable exhibits a positive effect or has a unidirectional relationship. It simply indicates that an increase in customer trust is bound to boost satisfaction. Similarly, [25] also stated that trust affects overall satisfaction.

Commitment is one of the most essential and central concepts for understanding the strength of customer relationships. It is important for measuring customer loyalty and predicting repeat buying behavior. The analysis results show that trust affects commitment because the t-value is TRàCM: 8.73 > 1.64 (real level 10%). Therefore, hypothesis 11 is accepted. This implies that every increase in confidence by one unit tends to boost commitment by 0.46 units.

This is in line with the theory of trust and commitment proposed by [20], which states that these variables are important in building a successful relationship. They further stated that there is a positive correlation between the two variables. Ref. [54] also stated that trust and commitment were positively correlated. Ref. [68] obtained similar results in relational marketing. Suppliers need to be trusted by retail stall owners or managers to ensure that they are highly committed to business dealings. In the end, it tends to boost their loyalty. It is important to note that suppliers in the FMCG retail industry usually maintain the trust of their customers in B2B.

The results show that the satisfaction variable affects commitment because the t-value is SFàCM: 8 > 1.64 (real level 10%). Therefore, hypothesis 12 is accepted. This signifies that each increase in satisfaction by one unit tends to boost commitment by 0.38 units. In this research, commitment is a psychological condition that describes the relationship between retail stalls and their suppliers. It affects their decision to either continue being loyal customers or go against their suppliers. The effect of satisfaction on commitment is because the services and products provided by the supplier suit their needs. This ultimately increases the retail stall owners' or managers' commitment to the supplier. Similarly, assuming these stalls are satisfied, they are bound to be more committed to maintaining their relationships with suppliers and continuously purchasing wholesale merchandise.

These are in line with the research carried out by [85], which states that satisfaction positively affects commitment. Ref. [37] also stated that this variable positively affects

commitment. This is because a higher level of customer satisfaction is bound to affect repeated purchase behavior.

### 4.3.5. The Effect of Commitment, Trust, and Satisfaction on Loyalty

The results of the analysis show that commitment affects loyalty because the t-value is CMàLY: 4.01 > 1.64 (real level 10%). Therefore, hypothesis 13 is accepted. This implies that every increase in commitment by one unit is bound to boost loyalty by 0.3 units.

This research also examined the effect of commitment on retail stall loyalty and reported a positive correlation. These results are consistent with the trust commitment theory proposed by [20] and relational marketing. Ref. [19] also reported that commitment significantly and positively affects customer loyalty. Similarly, [85] stated that commitment affects loyalty. Ref. [5] confirmed that this variable is an important driver of customer loyalty in the service industry.

To develop commitment, suppliers should render reliable services, such as the timely delivery of goods and taking the right orders. On the other hand, they also need to be able to provide varying quality goods worthy of sale and competitive pricing, in accordance with the needs of the stalls. It is paramount to create an application that is easy to understand and use.

The analysis shows that the trust variable does not affect loyalty, with a t-value of −1.19. This implies that the t-value is <1.64 (real level 10%). Therefore, hypothesis 14 is rejected. Interestingly, this research reported an insignificant effect between trust and loyalty. Trust is the key to building cordial relationships between suppliers and retail stalls, but it does not necessarily lead to loyalty. Another relationship foundation or basis is needed for satisfaction and commitment to boost loyalty.

These results align with the research by [5], which states that trust in suppliers does not affect purchase intentions or customer loyalty. However, some literature on relational marketing states that this variable leads to customer loyalty. Although, it has been proven by several preliminary research studies [19,22,25,32,75,89,93].

In this new digital transformation era, of course, building loyalty through relational marketing is no longer enough to develop trust. Rather, there is also a need for satisfaction and commitment. Retail stall owners or managers believe in the ability of suppliers to deliver wholesale purchases or merchandise, as well as diverse products easily ordered through the designed applications. However, it is not enough to boost loyalty. It takes a sense of satisfaction in retail stalls to become loyal to suppliers.

Based on these results, the satisfaction variable affects loyalty, with a t-value of 7.33. This simply denotes that the t-value is >1.64 (real level 10%). Therefore hypothesis 15 is accepted. These results indicate that satisfaction affects loyalty. The more retail stalls are satisfied with the services or types of products delivered by suppliers through the ordering applications, the greater the loyalty associated with meeting the wholesale needs of their merchandise. If the retail stalls are satisfied, they can decide to become loyal customers by continuing to make repurchases from the supplier.

These results are in line with various research that satisfaction positively affects customer loyalty [5,6,19,22,24,25,31,75]. Similarly, customer satisfaction affects loyalty, which has been widely proven in the literature entitled business-to-consumer relationships [10,23,28,32,44,52,82,89,91,105].

Ref. [92] stated that satisfaction has an insignificant effect on customer loyalty. Ref. [85] also reported that satisfaction indirectly impacts loyalty. However, customer satisfaction affects loyalty through the commitment relationship path. This research shows that the higher the perceived level of retail stall owners or managers' satisfaction, the greater their loyalty to the suppliers, and vice versa. Meeting the appropriate expectations suppliers give will result in satisfaction, affecting loyalty.

### 4.4. Indirect Effect

The indirect effect of loyalty was realized by analyzing the mediating role of commitment, trust, and satisfaction on retail service and website qualities, as well as merchandising on traditional retail stalls. Based on the results in Table 8, it is evident that the mediating effect of satisfaction on retail service quality, merchandising, and website quality is 85.8%, 67.9%, and 67.4%, respectively. Moreover, trust also plays a huge role of 78.7% and 56.4 in retail services and website qualities, respectively. The mediating variable, commitment, plays a large role of approximately 79.9% in retail service quality but has an insignificant impact on website quality.

**Table 8.** Significance of the satisfaction, trust, and commitment roles in the indirect and total effects on loyalty.

| No | Variable | Direct Effect | Indirect Effect | | | Total Effect | % Indirect Effect |
|----|----------|---------------|------------------|-------|------------|--------------|-------------------|
| | | | **Satisfaction** | **Trust** | **Commitment** | | |
| 1 | Service Quality | 0.03 | 0.181 | | | 0.211 | 85.8 |
| | Service Quality | 0.03 | | 0.111 | | 0.141 | 78.7 |
| | Service Quality | 0.03 | | | 0.119 | 0.149 | 79.9 |
| 2 | Merchandising | 0.06 | 0.127 | | | 0.187 | 67.9 |
| 3 | Web Quality | 0.13 | 0.269 | | | 0.399 | 67.4 |
| | Web Quality | 0.13 | | 0.168 | | 0.298 | 56.4 |
| | Web Quality | 0.13 | | | 0.127 | 0.257 | 49.4 |

Based on the results of the indirect effect analysis, the impact of the mediating variables, namely satisfaction, trust, and commitment to loyalty, are as follows:

1. In service quality, satisfaction is the biggest mediating role that affects loyalty, with indirect and total effects of 0.181 and 0.211, respectively.
2. Satisfaction significantly impacts merchandising, which affects loyalty, with indirect and total effects of 0.127 and 0.187, respectively.
3. The role of the most significant mediating variable on website quality that affects loyalty is satisfaction, with indirect and total effects of 0.269 and 0.399, respectively.

#### 4.4.1. The Effect of Service Quality on Loyalty through Trust and Satisfaction (SQ→TR → SF→LY)

Based on the results of the indirect effect analysis, retail service quality (SQ) significantly affects loyalty (LY) through trust (TR) and satisfaction (SF) by 0.956. Moreover, suppliers must partner with retail stall owners or managers to render good-quality services by supporting rapid delivery policies. This helps boost the payment method, either cash on delivery or with pay latter It affects one's confidence in placing goods or orders with suppliers, thereby boosting their satisfaction and loyalty to suppliers.

#### 4.4.2. The Effect of Merchandising on Loyalty through Satisfaction (MD→SF→LY)

Based on the results of the indirect effect analysis, merchandising (MD) has a significant impact on loyalty (LY) through satisfaction (SF) by 0.1512. These findings show that this variable significantly affects retail stalls' loyalty to suppliers. Therefore, a stronger relationship foundation is needed, namely satisfaction. Suppose a retail stall is about to place an order for goods through the application. In that case, it sees the items uploaded on the application, complete with various products, and the availability of the goods. It is bound to satisfy the retail stalls to shop with suppliers at competitive prices, thereby boosting their loyalty.

#### 4.4.3. The Effect of Website Quality on Loyalty through Trust and Satisfaction (WQ→TR → SF→LY)

Based on the results of the indirect effect analysis, website quality (WQ) has a significant impact on loyalty (LY) through trust (TR) and satisfaction (SF) by 0.1129. Retail stalls in

B2B deal with suppliers directly; however, in this digital transformation era, merchandise is purchased using the ordering application. These stalls feel satisfied because the supplier selects the goods through the application, places an order, and has them delivered. They feel practically efficient in terms of time, effort, and cost. If the retail stalls are satisfied, they will become loyal to the supplier. Those satisfied with the ordering application will make repurchases by using the app due to its benefits, such as not having to close the stalls.

## 5. Conclusions

Based on the discussion of the results of the research on the retail stall loyalty model to suppliers in the era of digital transformation in the Indonesian FMCG retail industry, it can be concluded that the determinants that directly affect the loyalty of traditional retail stalls to suppliers in the digital transformation era are merchandising factors, website quality factors, commitment, and satisfaction.

Merchandising factors from suppliers are mainly related to a complete and appropriate product assortment for traditional retail stalls, competitive prices, stock availability, and quality products. The website quality factor is mainly related to ordering applications that are easy to understand and easy to use by traditional retail stalls to order wholesale merchandise for suppliers. The commitment factor and satisfaction factor for important for suppliers. The results in this study indicate that the satisfaction variable is the most important antecedent for traditional retail stall loyalty to suppliers, and the results of the study also found that the merchandising factor is the main determinant of satisfaction.

The findings in this study confirm the mediating role for satisfaction, trust, and commitment in traditional retail stall loyalty to suppliers. Service quality does not have a significant effect on customer loyalty directly, but service quality has a significant influence on loyalty through the role of satisfaction and commitment. The effect of service quality on loyalty through satisfaction becomes stronger than the effect of service quality on loyalty through commitment. The effect of merchandising on loyalty becomes stronger through the role of satisfaction compared to the direct effect of merchandising on loyalty. The effect of website quality on loyalty is greater than the role of satisfaction, compared to the direct effect of website quality on loyalty. Satisfaction has a large influence on loyalty, but the effect of satisfaction on loyalty becomes stronger if through the role mediation of commitment. The results also show that trust does not have a direct effect on loyalty, but trust has an influence on loyalty through the roles of commitment and satisfaction.

The loyalty model in this study shows that the establishment of traditional retail stall loyalty to suppliers in the digital transformation era in Indonesia is demonstrated by the existence of quality websites from suppliers with secure ordering applications for transactions; applications that provide information that can be trusted, easy to understand, and easy to use by customers who shop at, traditional retail stalls for ordering wholesale merchandise. Website quality with a good ordering application is used for wholesale retail stall shopping with variable merchandising of merchandise from suppliers with competitive price conditions, sufficient stock availability, complete goods, and quality goods. Good supplier service quality supports problem solving during interactions, both in terms of delivery reliability and the ease of payment. The existence of a free shipping policy and a payment policy via Fintech is also very much needed for suppliers. This process ultimately forms a quality relationship in traditional retail stalls to feel trust with suppliers and foster a sense of satisfaction in traditional retail stalls with a commitment to be loyal to suppliers.

Studies on the loyalty of B2B relationships in the FMCG retail industry in the era of digital transformation are still limited. Therefore, this study contributed to the B2B marketing literature by focusing on service quality and merchandising, as well as website and relationship quality, thereby filling gaps in the literature. It examined B2B consumer behavior in traditional FMCG retail channels and evaluated the factors that influence traditional loyalty to suppliers in the era of digital transformation.

Its contribution to the academic literature is in sixfold. First, it proposed a new conceptual framework that highlights the B2B relationship between suppliers and traditional retail stalls. Verifying the conceptual framework provides empirical evidence for the relationship between merchandising, website, and service qualities, and relationship quality factors on loyalty. Overall, the result showed that merchandising and website quality have a significant direct effect on increasing traditional retail stall loyalty to suppliers. Meanwhile, service quality does not directly affect traditional retail stall loyalty to suppliers. Merchandising dimensions in the form of product diversity from suppliers, the availability of goods, product quality, and pricing strategies for supplier applications are important points that need to be considered because they affect the loyalty of traditional retail stalls.

Second, this research expanded the literature on digital transformation in relation to the relationship between service quality and the loyalty of traditional retail stalls to suppliers. The results of the study show that service quality has no significant effect on retail shop loyalty to suppliers. In the era of offline to online digital transformation in traditional retail stalls, the role of supplier personnel or employees interacting with retail stall owners is starting to diminish, along with how to order traditional retail stalls to suppliers who already use ordering applications. This is, of course, interesting because the research results show that service quality has no significant effect on loyalty, but website quality has a significant effect on the loyalty of traditional retail stalls for suppliers.

Third, this study expanded the literature on digital transformation by deepening knowledge about how website quality positively impacts traditional retail stall loyalty to suppliers. In the current era of digital transformation, the variable quality of the website is also a point that has enough attention because it is the main tool for placing orders and is also used as a means of supplier information to traditional retail stalls. The quality of the website designed for ordering goods from suppliers also needs to easy to use, understand, and provide clear information for traditional retail stalls.

Fourth, this study also contributes to the literature on commitment, trust, and satisfaction in FMCG retail. From the analysis on the behavior of B2B relationships between supplier companies and traditional retail stalls, it is known that commitment and satisfaction have a significant effect on traditional retail stall loyalty to suppliers. Furthermore, trust has no significant effect on traditional retail stall loyalty to suppliers. It is important to note that the trust variable is essential when dealing with traditional retail stalls in business with suppliers to earn their loyalty.

Fifth, the results showed that the role of the mediating variable satisfaction is very large on retail service quality, merchandising, and website quality on retail shop loyalty. For the mediating role, the trust variable is also very large on retail service quality and website quality on traditional retail stall loyalty to suppliers. Meanwhile, for the mediation variable role, commitment is very large on retail service quality for traditional retail stall loyalty, and its role is not large enough on website quality for retail traditional loyalty. This implies that the role of trust, commitment, and satisfaction in the formation of traditional retail stall loyalty to suppliers is very large and has an indirect positive impact on the variables of service quality, merchandising, and website quality.

Sixth, the findings of this study can be used both locally and internationally. Indonesia is a country with a population of 3.6 million traditional FMCG retail stalls, in the context of MSMEs. The country's struggle from traditional to digital services in business deserves to be noticed and appreciated.

*5.1. Theoretical and Managerial Implications*

This research focuses on Indonesia's fast-moving consumer goods retail industry. It also observed the relationship between supplier companies and their customers, namely traditional retail stalls.

The findings obtained from descriptive analysis and data processing with structural equation modeling (SEM) have several managerial implications in Indonesia's traditional FMCG retail industry. The results of the study show that merchandising, website quality,

commitment, and satisfaction have a significant effect on the loyalty of traditional retail stalls to suppliers. Meanwhile, service quality and trust have no significant effect on the loyalty of traditional retail stalls. Satisfaction and website quality variables are the strongest determinants that affect traditional retail stall loyalty to suppliers. Meanwhile, the merchandising variable is the strongest determinant that influences traditional retail stall satisfaction with suppliers.

Suppliers need to improve their ability to provide proper merchandising in conjunction with traditional retail stalls. In addition, procuring and fulfilling products by suppliers in terms of quantity, variety, quality and appropriate price is essential for fast-moving consumer goods retail businesses. In relation to traditional retail stall partners, suppliers must pay attention to diversity, the availability of merchandise, good product quality, and competitive prices.

The supplier's ability to provide a good quality website for ordering goods from retail stalls needs to be developed. In improving website quality, there is a need to pay attention to usability factors, which are easy to understand and used in placing orders to suppliers. The ordering application can also present quality information, product display, prices, and reliable promotions that are easy to understand, relevant, and accurate. Furthermore, service interaction quality should be emphasized. This makes retail stalls feel safe when ordering goods from suppliers and during transactions, including selecting merchandise and paying through applications. Then, its appearance for ordering goods at suppliers, which is overall good, encourages retail stalls to make wholesale purchases due to the easy and practical shopping experience. The biggest customers from these suppliers are traditional retail stalls whose average age does not belong to the millennials and, therefore, need a user-friendly ordering application.

For suppliers to build relationships to retain their customers successfully, they need to pay attention to the trust, commitment, and satisfaction of their retail stalls. The role and support of top management suppliers are required in making strategically important decisions in these companies that are closely related to the retail side of the service, merchandising, and website quality to increase the retail stalls' loyalty. Support from the principal stakeholders who are the source of the supplier's merchandise should also be noted. The merchandising aspect will improve if the product and purchase prices are supported. Meanwhile, the support from top management at the supplier companies in providing policies can improve reliable services and help retail stall partners. For example, these are related to ordering goods through applications, making it easier to transact wholesale shopping. Retail stall owners or managers need not need to close their shops when they go shopping. The wholesale shopping payment system policy can be cash on delivery, payment with a digital wallet or pay letter service, a tempo payment system, and an on-time delivery policy (same-day service) with free shipping. Subsequently, it can form and increase retail stall loyalty, especially in the new digital transformation era.

*5.2. Limitations and Suggestions for Further Studies*

This research is part of a preliminary study in developing countries, especially in Indonesia, which empirically examines the loyalty of traditional retail shops to suppliers in the digital transformation era. Loyalty is the most important asset for the company's survival, competitive advantage, and business continuity. The main focus of this research is to unravel and analyze the factors that influence the loyalty of retail stalls to FMCG retail suppliers in the era of digital transformation based on the buyer–seller relationship theory and relationship marketing. This study also analyzes the role of trust, commitment, and satisfaction in retail service quality, merchandising, and website quality on loyalty. These include retail service, merchandising, website, and relationship quality. According to [18], some factors are used as variables in the success of relationship marketing-based strategies that can be applied in further research. This includes CRM (customer relationship management) by implementing a data-based program to manage the relationship between suppliers and retail stalls efficiently and effectively.

Although this research can provide some contributions, there are still limitations. The limitation associated with this study is the conduction of sampling in traditional retail stalls that are already members of the supplier. Therefore, a further study is recommended for samples from those selected at random. It is also limited to retail stalls that already use applications to order wholesale goods to suppliers; hence, a further study needs to examine the factors that influence retail stalls' adoptability of technology in wholesale ordering systems through applications. It is important to note that this study focused on B2B relationships in the FMCG retail industry, but further study could be conducted in different industries or sectors to compare the findings.

**Author Contributions:** Conceptualization, M.M., H.H., R.N. and E.Z.Y.; methodology, M.M., H.H., R.N. and E.Z.Y.; software, M.M.; validation, H.H., R.N. and E.Z.Y.; formal analysis, M.M., H.H., R.N. and E.Z.Y.; investigation, M.M.; resources, M.M., H.H., R.N. and E.Z.Y.; data curation, M.M., H.H., R.N. and E.Z.Y.; writing—original draft preparation, M.M., H.H., R.N. and E.Z.Y.; writing—review and editing, M.M., H.H., R.N. and E.Z.Y.; visualization, M.M.; supervision, H.H, R.N. and E.Z.Y.; project administration, M.M.; funding acquisition, M.M., H.H., R.N. and E.Z.Y. All authors have read and agreed to the published version of the manuscript.

**Funding:** This research received no external funding.

**Institutional Review Board Statement:** Not applicable.

**Informed Consent Statement:** Not applicable.

**Data Availability Statement:** All the data have been included in the manuscript.

**Conflicts of Interest:** The authors declared no conflict of interest.

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
