# Peer review of "The Unraveling Loyalty Model of Traditional Retail to Suppliers for Business Sustainability in the Digital Transformation Era: Insight from MSMEs in Indonesia"

_sustainability, doi:10.3390/su15032827_

Round 1

Reviewer 1 Report

Dear authors 

The article follows an interesting idea with a well-design structure. I have some suggestions for this paper. 

1. You are providing a lot of information. I would like to suggest you add a section after the literature review to directly explain the research's importance and contributions. A separated section or sub-section. It helps the reader to skip many lines to read your paper easier. 

2. Indonesia’s FMCG MSMEs as a keyword. Please delete Indonesia and put the full name of FMCG MSME instead. 

3. Add a flowchart, figure, or something else to clarify what you are going to do in the paper after the introduction part. It helps a lot of readers. 

4. The references look not really new. Please update your study with some newer references. 

Author Response

Dear Reviewers

Thank you very much for the excellent suggestions and input which really helped us in the revision of this research article. 

Thank you for suggestions to add a section after the literature review to directly explain the importance of the research and contribution. Suggestions related to keywors and also the flow to clarify the stages of the article after introduction chapter. This is very useful input ti help make this article easier to read.

For reference we have also added and update it. There are 58 refernces that we have updated for the last 5 years.

Thank you.

Reviewer 2 Report

·        Introduction.

The new retail hybrid concept is used but not defined (43)

The entire second paragraph of the introduction (43-54) lacks bibliographical references.

The phrasing of paragraph (59-62) is somewhat confusing.

The statement "Service quality is the main source of competitive advantage in the market." (71-72) has no bibliographical reference.

The statement "Market supply factors in tangible attributes called goods also affect the successful strategy employed in relationship marketing." (77-78) has no bibliographical reference.

The terms B2B and B2C are not defined before and are, however, used in the text.

The entire paragraph between lines 77 and 83, lacks references

The statement "The relationship between merchandising, customer satisfaction, and loyalty has not been stated in the B2B marketing literature", from which the study gap is derived, does not have a reference author.

The phrase "The success of relationship marketing depends on the benefits it offers to the customers" (112-113), lacks a reference author.

·        Literature Reviews.

The paragraphs included in lines (149-162) lack reference authors.

The phrases "Developing customer loyalty is the main goal of a company. It provides various benefits for the organization in the long run. However, in the context of B2B relationships, these advantages benefit both partners in the business." (184-186) lack a reference author.

The phrases "Furthermore, customer loyalty is important in building and developing long-term, mutually beneficial relationships between suppliers and customers." (191-193) lack a reference author.

The phrases "Making purchases from other lines of products or services, referrals involving the recommendation of suppliers to friends or colleagues, and retention or resistance to negative impacts affects this company." (195-197) lack a reference author

Lines 219-220 refer to the RSQS scale and the five dimensions of retail service. However, it is not clear in which part of the study this scale is put into practice.

The sentences "Internet-based retailers should be able to meet the competitive prices uploaded on the website" (233-234), and Certain preliminary studies have also proven that selling quality products at low prices is not enough to satisfy longtime loyal customers" (237-238) lack references. In addition, it is indicated that there are certain theories, without indicating what they are.

The reference of line 243 is not correctly made.

The statements on lines 248-250; and 254-256, lacks reference authors.

The paragraph between lines 258-265, with the exception of a specific phrase, lacks reference authors.

The paragraph between lines 288-296, lacks reference authors.

The paragraph between lines 309-313, lacks reference authors.

·        Hypothesis

It would be interesting to know, in the development of the hypotheses, what is the reason why variables as broad as quality of service or merchandising are later combined with a variable as specific as the quality of the website.

·        Methodology

It would be interesting to know the criteria by which it was decided that the study was quantitative.

All section 3.1 lacks reference authors. It would be important to place authors who endorse the statements made regarding the data analysis technique, their preference over the use of linear equations, as well as the data collection technique.

It is indicated that it started from a population of more than 60,000 elements and concluded with a sample of 500 elements. This, at first glance, is clearly insufficient. However, the existence of certain validation criteria for the informants is also pointed out. These criteria, however, are not detailed. The suggestion is to demonstrate the sample calculation and detail the inclusion criteria of the informants in the sample.

·        Discussion

Demographic percentages of the sample, especially age and gender, should reflect the actual percentages of the population from which the sample was drawn.

·        Conclusions

The term "kiosk" is only used for the first time in the conclusion. It is important to know if it is a new concept, because if it is the case, it should have been dealt with in the theoretical framework.

It is indicated that the use of structural equations has certain managerial implications, however the relationship is not made adequately and the idea is not clear.

The conclusions indicate that the relevance of the study of the web page as a variable is given by the age of the managers. However, throughout the paper, especially in the theoretical framework, age is never taken as a study factor.

Author Response

Dear Reviewers.

Thank you very much the excellent suggestions and input which really helped us in the revision of this research article. We have revised this article based on informed feedback.

In the Introduction section. We have added relevant references, we have also added the purpose of the paper to be discussed, research problem, research questions and theoretical discussions and also research gap that justify the contribution of this research.

In the Literature review section. We have also revised it by adding relevant references. We reorganized and rearranged this section so that we have clear references in accordance with the directions from reviewer.

On the Hypothesis, we have also provided additional explanations and reasons why variables as broad of service quality or merchandising are combined with specific variables as website quality.

In The Methodology section, we have revised it according to the suggestions of the reviewer. We revised by completing references that support statements regarding data analysis techniques and data collection techniques. We also explain the sampling techniques and its criteria, including details on calculating the number of samples. 

Discussion section , we have also revised the demographic percentage of the sample, especially age and gender, which reflects the actual the population where the sample was taken.

In the Conclusion section, we have also revised according to the directions reviewer, including the managerial implications of this article.

Thank you.

Reviewer 3 Report

Hi Authors. 

Congratulations on your excellent contribution to the academic literature. 

I went through this article carefully, and my main concern is the reference for a few statements. According to the number of references I see in this article (86), I am pretty sure that all statements have enough support. On the other hand, the way they are written makes me think they are statements from the authors, not statements based on references. The "Literature Reviews" section needs to be carefully reviewed to avoid this possible misinterpretation of how the statements were written. I noticed the same in the "Introduction," and I will list all of them in more detail below.

Lines 043-047: Deserve a reference since it is an important statement.

Lines 051-053: Saying it requires "sales canvassers" is important and deserves a reference.

Lines 071-073: It isn't very clear. Is quality the main source of competitive advantage or a prereq? It seems that quality is expected (the baseline) for any business. Please, go through this again to be more precise. There is no reference as well.

Line 106-107: It needs to be clarified what website design quality means. What does that mean? Fewer clicks? Making it more explicit will be beneficial for this article.

The literature review needs to be clarified. The authors (it looks like) mix statements from the references and references from the authors. For example, lines 144-148 are pretty clear, but lines 149-162 seem to be the authors' personal opinion. The same happens in 171-174. You can also find this in 177-180. Lines 184-186 is one more example. Actually, a statement starts the section, but there is no reference to reinforce the it. The same with 191-194. Also happens in 195-197. One more in 211-212. Again in 217-219. 233-234. 237-238. 248-250. 254-256. 258-259. 262-265. 281-283. A big one in 287-296. Since the section refers to Literature Reviews, it deserves more references or at least makes clear all statements have a source. It will make this article stronger.

The same happens in Section 2.2. It starts with "In the past, multivariate regression methods and factor analysis were frequently used to detect interdependence in the research problem being examined". Who is stating this?

Lines 342-344: Is that possible to have more details from where did H2, H3 and H4 come from? The same with 360-361. On the other hand, lines 380-381 are pretty clear relating the hypotheses with the findings.

Lines 456-457: This statement "This research adopted a quantitative approach with questionnaires compiled and distributed to respondents using the personal self-management method.". Is that possible to have a description of this method? It is an important piece of the applied methodology.

Page 11 and 15: It seems a typo "Ritel Service Quality".

I want to raise this topic again: "website quality". This concept seems broad and can impact many perspectives of digital transformation. I would recommend adding more to this to clarify what you are mentioning.

Thank you and all the best.

Author Response

Dear Reviewers.

Thank you very much for the excellent suggestions and directions which really helped us in the revision of this research article. We have revised this article based on informed feedback.

In the Introduction section, we have added relevant references, we have also added the purpose of the paper to be discussed, research problems, research questions, and theoretical discussions and also research gap that justify the contribution of this research.

In the Literature Review section, we have carefully revised adding the relevant references to avoid misinterpretation of how the statement was written.

In the Hypothesis, we have also revised the detailed directions for how hypotheses are made from the existing literature.

In The Methodology section, we have revised it according the direction of the reviewer. We revise by completing references that support statements regarding data analysis techniques and data collections techniques. We also explain the sampling techniques its criteria, including details on calculating the number of samples.

Thank you.

Reviewer 4 Report

This article interestingly investigates the unravelling loyalty model of traditional retail’s to suppliers in the digital transformation era: insight from MSMEs in Indonesia. However, I have the following comments that need to be addressed:

1-          The introduction needs improvements; the objective of the paper should be discussed in the context of the discussion. This should be based on the research problem, questions, and theory discussions. The research problem and questions should be discussed in the introduction. Further, the research gap that justifies your contribution with theory underpinnings need to be discussed. It is necessary to discuss and mention how your research contributes to the strand literature. The introduction in its current format have excessive supportive literature on the account of its contribution and research gap.

2-          In your hypotheses, I suggest you that you frame hypotheses without mentioning positive or negative. At this stage, it is enough to mention significant rather than positive and negative.

3-          Some typos are there accordingly, proofreading is needed.  

Author Response

Dear Reviewer.

Thank you very much for the excellent suggestions and directions which really helped us in the revision of this research article. We have revised this article based on informed feedback.

We have added relevant references, we have also added the purpose of the paper to be discussed, research problems, research questions, and theoretical discussions and also research gaps that justify the contribution this research.

We have also revised according to suggestions from reviewers regarding the hypothesis by changing positive and negative to be significant.

For some typos, we have also corrected and revised them.

Thank you.  

Round 2

Reviewer 2 Report

I think the vast majority of suggestions have been taken into account. The references are sufficient, and the methodology has become clearer. I think, however, that they could still work on the conclusions, as they do not reflect the full depth of the findings.

Author Response

Dear Reviewer

Thank you very much for your comment and suggestion.  I ready make additional conclussions as follow ( attachment). 

Many thank
